# Comparison of Venous Blood Gas and Biochemical Parameters in Sunda Pangolin (*Manis javanica*) and Chinese Pangolin (*Manis pentadactyla*) before and after Isoflurane Anesthesia

**DOI:** 10.3390/ani13071162

**Published:** 2023-03-25

**Authors:** Fuyu An, Hongmei Yan, Xuelin Xu, Yingjie Kuang, Xianghe Wang, Zhidong Zhang, Zhenyu Ren, Jiejian Zou, Fanghui Hou, Kai Wang, Yan Hua

**Affiliations:** 1Guangdong Provincial Key Laboratory of Silviculture, Protection and Utilization, Guangdong Academy of Forestry, Guangzhou 510520, China; 2College of Animal Science and Technology, Zhongkai University of Agriculture and Engineering, Guangzhou 510550, China; 3Guangdong Wildlife Rescue Monitoring Center, Guangzhou 510520, China; 4Pangolin Conservation Research Center of National Forestry and Grassland Administration, Guangzhou 510520, China

**Keywords:** *Manis javanica*, *Manis pentadactyla*, pangolin, blood gas, respiratory, anesthesia

## Abstract

**Simple Summary:**

The Sunda pangolin (*Manis javanica*) and Chinese pangolin (*Manis pentadactyla*) are critically endangered mammals, and there is limited information on their blood physiological reference values in the literature. Venous blood was collected from wild healthy Sunda pangolins (*M. javanica*) and Chinese pangolins (*M. pentadactyla*) before and after isoflurane anesthesia was given, with the aim of establishing reference ranges of venous blood gas values for conscious and isoflurane-anesthetized pangolins. The results showed that there was a strong consistency in the blood gas index trends of the two pangolin species under waking and isoflurane anesthesia states. This study provides data to help understand the physiological status of healthy Sunda and Chinese pangolins for comparison with different life stages and disease states.

**Abstract:**

Venous blood gas analytes are commonly examined in animals, and the results may be important when evaluating the overall health status of an animal. Pangolins are critically endangered mammals, and there is limited information on their physiological reference values in the literature. The aim of this study was to analyze venous blood gas and biochemical parameters before and during isoflurane anesthesia in wild healthy Sunda and Chinese pangolins. The results obtained showed that the blood gas index trends of the two pangolin species before and after isoflurane anesthesia were the same. After anesthesia, the partial pressure of carbon dioxide (pCO_2_), partial pressure of oxygen (pO_2_), total carbon dioxide (CO_2_), mean blood bicarbonate (HCO_3_^−^), extracellular fluid compartment (BEecf) base excess and the mean blood glucose (Glu) levels of both pangolin species showed a significant increase compared to the pre-anesthesia period. In contrast, the mean blood potassium (K^+^), lactate (Lac) and mean blood pH levels were significantly lower. No significant differences in the mean blood sodium (Na^+^) or blood ionized calcium (iCa) levels were observed during anesthesia. This study is important for future comparisons and understanding the health status of this endangered species.

## 1. Introduction

The Sunda pangolin (*Manis javanica*) and Chinese pangolin (*Manis pentadactyla*) are both listed as endangered species [1,2,3] and are among the most trafficked mammals in the world [4]. All pangolin species are classified as a National Level I Protected Animals in China [5] and are listed as Critically Endangered on the International Union for Conservation of Nature (IUCN) Red List of Threatened Species [6,7,8]. Habitat loss, hunting for international trade, and a low reproductive rate have led to a dramatic decline in wild pangolin populations [9,10,11,12,13]. Accordingly, there is an urgent need to consider ex situ conservation as part of a holistic approach to pangolin conservation. A number of existing pangolin rescue centers in countries within their range, including China [14], India [15], Singapore [16], Vietnam [17] and South Africa [18], rescue and rehabilitate confiscated pangolins prior to their release back to the wild. Normal physiological characteristics and common health concerns have rarely been studied in pangolins compared to many other well-known wild animals [19,20,21,22], resulting in challenges for rehabilitation centers and conservation breeding programs in the pangolin range.

Hematological parameters are important physiological indicators for animals, and blood biochemical studies are important for assessing the health and physiological state of organs in animals. Blood gas analysis can provide valuable information about the cardiopulmonary and acid-base status of a critically ill veterinary patient [23,24]. Several studies have evaluated the hematological and serum biochemical parameters in the Sunda pangolin and Chinese pangolin from various regions of the world [14,16,17,25,26], but blood gas analysis results from these two pangolin species have not been reported. To the best of our knowledge, there is only one report of blood gas values in pangolins, but these values were described in the white-bellied pangolin (*Phataginus tricuspis*) [27]. Pangolins seized from the wildlife trade for rehabilitation may be dehydrated and malnourished [18]. Venous blood gas and biochemical analyses can effectively assess the health status of rescued wild pangolins and guide appropriate treatment in veterinary practice.

The aim of this study was to analyze venous blood gas and biochemical parameters in rescued wild Sunda pangolins (*M. javanica*) and Chinese pangolins (*M. pentadactyla*), to establish normal ranges for venous blood gas, pH, electrolyte, blood ionized calcium (iCa) and Lac parameters in awake and anesthetized animals and improve knowledge about the physiology and anesthetic management of these two pangolin species.

## 2. Materials and Methods

### 2.1. Animal Population

This study included rescued pangolins from different counties in Guangdong Province who were transported to the Guangdong Provincial Wildlife Rescue monitoring center by local government officers and the public from 2019 to 2022. Each animal was isolated and quarantined for at least 1 month. The pangolins were treated for injuries or medical problems, if any, until apparently good health was achieved. Pangolins that were active, had a good appetite, produced normal feces, were free of parasites or traumatic injuries, and had a stable weight were considered to be in good health [14].

Twenty-eight adult Chinese pangolins (*M. pentadactyla*) and twenty-six adult Sunda pangolins (*M. javanica*) were analyzed in this study. The adult Chinese pangolin population included 14 males and 14 females, with weights ranging from 3.1–5.96 kg. The adult Sunda pangolin population included 13 males and 13 females, with weights ranging from 2.94–8.87 kg. The total body length and tail length of each pangolin were measured to accurately assess age (Table 1). Detailed data on the weight, total length, and tail length of each pangolin can be found in the Appendix A.

### 2.2. Sample Collection and Processing

All blood samples (1 mL) were drawn from the medial caudal vein using a 1-cc preheparinized syringe and 22-gauge needle. For the awake (T1) pangolins, blood was drawn manually from the tail ventral vein. After the first (T1) sample collection, each pangolin was placed in an inhalation chamber to induce anesthesia, and 5% isoflurane in a 2 L flow of oxygen/min was circulated in the chamber until the animal’s muscles had relaxed. Following induction, anesthesia was maintained with 2% isoflurane in a 1.2 L flow of oxygen/min delivered via a small facemask and a Mapleson D nonrebreathing circuit (Superstar Medical Equipment, Nanjing, DM6A, China). Body temperature was measured with a rectal thermistor probe (Mindray, Guangzhou, uMEC12Vet, China) and maintained with a warm blanket and heat packs. Oxygen saturation was measured by pulse oximetry (Mindray, Guangzhou, uMEC12Vet, China), and vital signs were monitored every 3 min with a stethoscope and a Doppler ultrasound machine (Parks, 811-B, Portland, OR, USA). Additionally, the duration of anesthesia, age, weight, and sex of each pangolin were recorded. Ten minutes after the induction of anesthesia, the T2 samples were collected at the same location as the T1 samples. Following the T2 sample collection, the anesthetic gas was turned off, and the pangolins were provided oxygen via a facemask until purposeful movement was observed. Once awake, the pangolins were placed in a heated incubator.

Blood gas, pH, and biochemical analyses (GEM Premier 3000, Instrumentation Laboratory Inc., Lexington, MA, USA) were performed immediately after sample collection. The tested analytes included pH, partial pressure of carbon dioxide (pCO_2_), partial pressure of oxygen (pO_2_), total carbon dioxide (TCO_2_), bicarbonate (HCO_3_^−^), base excess (BE), oxygen saturation (sO_2_), lactate (Lac), sodium (Na), potassium (K), total carbon dioxide (TCO_2_), anion gap, ionized calcium, glucose, blood urea nitrogen (BUN), and hematocrit (Hct). Samples were analyzed at 37–38 °C by default, and variables were not temperature-corrected at the time of analysis. Uncorrected values for pH, pCO_2_, pO_2_ and HCO_3_^−^ were then temperature corrected using the correction formulae supplied by the manufacturer of the analyzer.

### 2.3. Statistical Analysis

Statistical analysis was performed with SPSS software version 25. A paired Student’s *t* test was used to reveal the statistical significance of the blood gas parameters between awake and anesthetized states in the two pangolin species. The normal distribution of the data was determined using the Shapiro-Wilk test, and differences between normally distributed data were considered significant when *p* < 0.05. For each species, the Wilcoxon signed-rank test was performed to identify associations of weight, length, and body temperature with the measured blood parameters. A Wilcoxon signed-rank test was used to compare the medians and ranges of all test values between the T1 and T2 samples (*p* < 0.05). Pairwise comparisons were performed to determine the impacts of anesthesia on venous blood gas analytes before and after anesthesia in the two pangolin species.

## 3. Results

The median time under anesthesia was 20 min (range, 15 to 32 min). There were no complications observed during anesthesia or recovery. Descriptive statistics were calculated for venous blood gas analytes collected immediately from the two pangolin species before anesthesia and after 10 min of isoflurane anesthesia (Table 2 and Table 3). Detailed data can be found in the Appendix A.

The temperature-corrected pH values and pH-corrected iCa values obtained with a GEM Premier 3000 analyzer were not significantly different from the combined uncorrected values in the T1 (awake) and T2 (isoflurane anesthesia) samples (in both Chinese and Sunda pangolins) (*p* > 0.05). There were no statistical correlations between the blood parameters and weight, total length, tail length, and body temperature of the Sunda and Chinese pangolins (*p* > 0.05). For both pangolin species, the mean blood pCO_2_, mean blood pO_2_, mean blood CO_2_, mean blood HCO_3_^−^, mean blood BEecf base excess and mean blood glucose were significantly higher in T2 than in T1 (*p* < 0.05). In contrast, the mean blood K^+^, mean blood Lac and mean blood pH levels were significantly lower in T2 than in T1 (*p* < 0.05). No significant difference was observed in the mean blood Na^+^, or mean iCa levels from the induction of anesthesia to the time immediately prior to anesthetic gas shutoff (*p* > 0.05). No complications occurred during or after anesthesia in any animal.

## 4. Discussion

This study provides the first blood gas reference ranges for wild Sunda pangolins (*M. javanica*) and Chinese pangolins (*M. pentadactyla*). Compared to previous studies [25,27], this is one of the only reports describing the blood gas values of two pangolin species present throughout Asia while under isoflurane anesthesia. Although no studies have reported baseline blood gas references between two or more pangolin populations, this study provides a reference for future comparisons of differences in blood gas values between pangolin species. Although the small sample size is a weakness of this study, these pangolins represent 100% of the healthy population rescued and restored by the Guangdong Wildlife Monitoring and Rescue Center from 2019–2022.

At the Singapore Zoological Gardens (SZG) [16], the Save Vietnam’s Wildlife (SVW) [17] rescue center, and the Taipei Zoo [14], the ages of apparently healthy male and female pangolins are estimated by body weight. Obviously, this does not apply to pangolins that have just been rescued, as they have often been starved and dehydrated for long periods of time due to illegal capture and injuries. Weight loss is so common that it is impossible to accurately assess age by weight, even if some individuals appear healthy. For example, the rescued pangolins in the current study weighed less than 3 kg before rehabilitation, and after approximately one month of care, at which point they were considered to be in good health (as stated earlier: with a stable weight, etc.), they weighed more than 3 kg. Therefore, adult pangolins should be evaluated by not only their weight but also by more consistent physiological parameters, such as their total length. The body weight of a healthy male Chinese pangolin is more than 3.5 kg, and the body length is longer than 71 cm. The body weight of a healthy female Chinese pangolin is more than 3 kg, and the body length is longer than 67 cm. The body weight of a healthy male Sunda pangolin is more than 5 kg, and the body length is longer than 90 cm, and the body weight of a healthy female Sunda pangolin is more than 4 kg, and the body length is longer than 82 cm. Pangolins fulfilling the above criteria were classified as ‘‘adults” (Table 1).

PCO_2_ is used to evaluate the respiratory acid–base balance. Respiratory acidosis or respiratory alkalosis occurs when the pCO_2_ level is too high or too low. In this study, an increase in the pCO_2_ level was observed in all pangolins after anesthesia with isoflurane. Anesthesia causes respiratory depression and a decrease in the respiratory rate [28]. Moreover, continuous oxygen supplementation during anesthesia [29] leads to an increase in pO_2_ (Table 1 and Table 2). When the respiratory rate is decreased, there is a resultant decrease in respiratory ventilation, leading to hypoventilation. In addition, relaxation of the intercostal muscles and a decreased tidal volume can contribute to reduced alveolar ventilation and an increased end-tidal CO_2_ concentration [30]. It was also found on endoscopy that the pangolins’ tongues tended to block the larynx during anesthesia, which further led to hypoventilation and an increased pCO_2_ level (Figure 1). For this phenomenon, the veterinarian will use an endoscope to first observe the pangolin’s larynx, and then pull out the tongue so that the airway is open to guarantee the smooth operation of anesthesia. Therefore, when mask anesthesia is administered to a pangolin, it is recommended that the tongue be pulled out so that the airway is open to avoid the above situation. Critical cutoffs have not been established for the species, but a pCO_2_ level >55 mmHg is considered to be representative of hypoventilation, on the basis of results in other species [27,30]. Although the pCO_2_ level was >55 mmHg in more than half of the pangolins after anesthesia induction, a blood pH level close to 7.35 was maintained by retaining bicarbonate. This indicates a type of autocompensation, which allowed the pangolins to tolerate high pCO_2_ levels.

Venous pH was slightly higher in the T1 samples than in the T2 samples in this study. Blood pH is dependent on a ventilatory component (pCO_2_) and a metabolic component (bicarbonate concentration) [31,32]. The decrease in blood pH levels observed in this study may have been a result of ventilation changes. According to the acid–base equation (carbonic acid/bicarbonate: CO_2_ + H_2_O → H_2_CO_3_ → HCO_3_^−^ + H^+^), hypoventilation allows excess CO_2_ to accumulate in the body, which further leads to an increase in H^+^ and HCO_3_^−^ in the blood. Although the magnitudes of the increases in H^+^ and HCO_3_^−^ were not large and, as mentioned above, were most likely related to autocompensation, these results need to be further verified.

Blood lactic acid is a product of anaerobic glycolysis and can directly reflect tissue hypoperfusion and hypoxic conditions [33]. Lactate values in dogs [34] do not exceed 2 mmol/L [31], but both the T1 and T2 samples from the Sunda and Chinese pangolins in this study showed lactate values above the median of 5 mmol/L; these values were even higher than those reported in the white-bellied pangolin (*P. tricuspis*) (a median of 3.7 mmol/L) [29]. During anesthesia, the muscles of the whole body relax, and the lactate level decreases. Therefore, the elevated lactate levels in these pangolins most likely resulted from the struggle against manual restraint before anesthesia [29]. Relatedly, artificial restraint may increase the stress level in an animal and trigger an increase in the blood glucose level. For example, glucose can increase artifactually in rabbits subjected to stressful situations [35].

The K^+^ and Na^+^ concentrations in the blood samples of the Sunda and Chinese pangolins in this study were similar to the ranges reported by Jennifer et al. (2021) [17], Ahmad et al. (2018) [16], and Khatri-Chhetri et al. (2015) [26]. These data may indicate the normal ranges of K^+^ and Na^+^ in wild healthy Sunda and Chinese pangolins. However, this still needs to be further verified with data from wild pangolins. The decreases in serum K^+^ and Na^+^ concentrations during anesthesia may be due to a change in the intravascular volume associated with vasodilation from inhalant administration [29,36,37] or to an increase in the glucose level due to exposure to a stressful situation, which can cause intracellular fluid to flow outside the cell, affecting serum Na^+^ and K^+^ concentrations [38,39]. The clinical relevance of this finding is uncertain.

Since the solubility of gases in blood is affected by temperature [40], the blood gas values measured in animals with lower body temperatures were corrected for temperature by a blood gas analyzer [22,41,42]. Therefore, these values may not represent the physiological values in the animals themselves. Instead, it may be more appropriate to correct the temperature detected by the blood gas analyzer for the body temperature of the measured animal. Pangolins have a lower body temperature than most mammals [43,44]. When detecting blood gases in the pangolin, the test results were corrected by temperature to make the test values more representative.

The results showed that the blood gas values in the two pangolin species were highly consistent, but whether these represent accurate physiological indices for the two pangolin species needs to be further verified in studies with larger samples and continuous monitoring times.

The main limitations of this study include uncertainty regarding the ages of the adult animals (estimated by weight), the lack of continuous surveillance, and the possibility of subclinical illness despite thorough veterinary examinations. The way in which the amount of inhaled oxygen and anesthetic should be effectively estimated to effectively evaluate the safety of isoflurane anesthesia in pangolins is a direction for future studies. Although oral endoscopy was performed for the pangolins in this study, no successful intubation or ventilation was performed in the anesthetized pangolins, and these procedures could be challenging.

## 5. Conclusions

This was the first study to analyze venous blood gas and biochemical parameters in wild healthy Sunda pangolins (*M. javanica*) and Chinese pangolins (*M. pentadactyla*). The aim was to establish reference ranges of venous blood gas values for the two pangolin species under waking and isoflurane anesthesia states. Despite the fact that the blood gas parameters reported in this study belong to rehabilitated healthy pangolins and consequently do not reflect the values of dehydrated and malnourished pangolins, they will serve as a useful reference for clinicians treating pangolins. This study is important for future comparisons and for understanding the health status of this endangered species.

## Figures and Tables

**Figure 1 animals-13-01162-f001:**
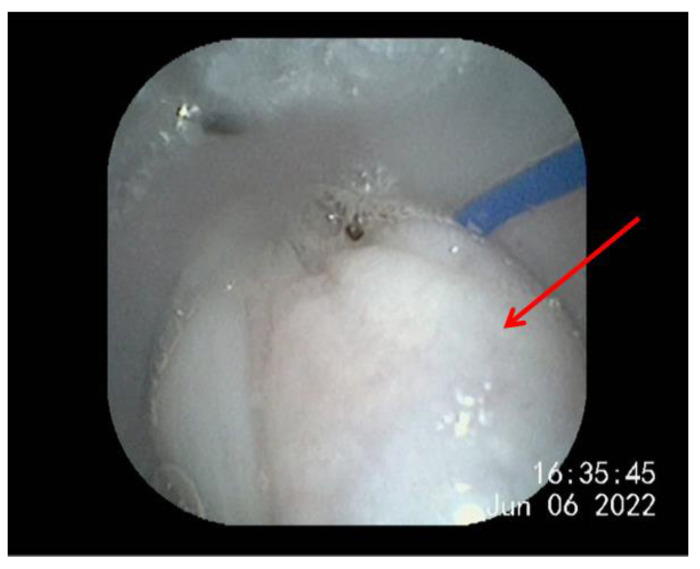
The conditions in the oral cavity of pangolins during isoflurane anaesthesia were observed by endoscope. The tongue was indicated by the arrow.

**Table 1 animals-13-01162-t001:** Mean, standard deviation (SD), median, and range (Min–Max) of weight, total length, and tail length data of adult pangolins (n*_M. P_* = 28, n*_M. J_* = 26).

Item	Sex	Median		Min		Max		SD		Mean	
		*M. P*	*M. J*	*M. P*	*M. J*	*M. P*	*M. J*	*M. P*	*M. J*	*M. P*	*M. J*
Weight	♀	3.43	4.82	3.1	3.71	5.96	5.93	0.73	0.58	3.6	4.86
	♂	4.32	6.66	3.29	5	5.94	8.87	0.71	1.47	4.38	6.84
Total length	♀	69	90	63.5	82	83	93	4.4	2.66	69.82	89.32
	♂	74.5	105	66	90	92	121	5.89	9.3	75.74	103.77
Tail length	♀	28	40	25	36	35	44.3	2.22	1.85	28.86	40.52
	♂	30.8	48	26	40	35	54.5	2.09	4.5	30.62	47.46

*M. P: Manis pentadactyla, M. J*
*: Manis javanica.*

**Table 2 animals-13-01162-t002:** Mean, standard deviation (SD), median, and range (Min–Max) of blood gas and lactate data provided in awake (T1) and isoflurane anaesthesia (T2) of Sunda pangolins (n_adult_ = 26).

Item	Unit		Awake			Isoflurane Anaesthesia			
		Median	Min	Max	Mean ± SD	Median	Min	Max	Mean ± SD
Body temperature	°C	33.1	32.4	33.9	33.142 ± 0.344	33.4	32.1	34.8	33.523 ± 0.636
H^+^	10^−7^ mol/L	4.624	3.02	8.128	5.098 ± 1.414	5.497	3.162	7.943	5.378 ± 1.249
pH	—	7.335	7.09	7.52	7.308 ± 0.115	7.26	7.1	7.5	7.281 ± 0.105
PO_2_	mmHg	95.5	23	144	96.231 ± 35.581	308.5	27.1	549	301.081 ± 143.662
PCO_2_	mmHg	52.5	36	109	62.615 ± 22.304	69.5	42	112	72.654 ± 16.68
pHt	—	7.39	7.15	7.58	7.362 ± 0.117	7.305	7.15	7.55	7.328 ± 0.107
PO_2_t	mmHg	75.5	18	125	78.038 ± 31.231	292.5	46	527	292.269 ± 132.21
PCO_2_t	mmHg	44	30	93	52.769 ± 18.688	61	35	93	62.577 ± 14.387
TCO_2_	mmol/L	31.95	19.7	58.3	32.396 ± 7.945	35	24.3	53.9	36.204 ± 6.786
Beecf	mmol/L	2.7	−9	29.6	4.181 ± 7.980	6.1	−6.6	26.6	7.238 ± 7.463
BE(B)	mmol/L	0.9	−8.8	20.4	1.858 ± 6.481	3.9	−7.9	19.6	4.119 ± 6.417
HCO_3_^−^	mmol/L	29.45	18.4	55.2	30.473 ± 7.526	32.8	22.3	51.4	33.965 ± 6.572
HCO_3_^−^(std)	mmol/L	25.65	18.1	38.3	26.081 ± 4.831	28.05	18.8	40.3	28.204 ± 5.005
K^+^	mmol/L	5.45	4.3	7.1	5.535 ± 0.696	4.1	2.9	5.4	4.085 ± 0.534
Na^+^	mmol/L	148	143	154	147.885 ± 3.103	145.5	140	153	145.808 ± 2.669
iCa2^+^	mmol/L	1.25	1.13	1.56	1.259 ± 0.1	1.22	1.08	1.53±	1.225 ± 0.084
Glu	mmol/L	4.35	3.1	10.2	4.873 ± 1.746	5.4	3.3	10.3	5.708 ± 1.574
Lac	mmol/L	8.2	1.3	13.5	7.738 ± 3.509	4.6	1.3	11.2	5.354 ± 3.223
nCa^2+^	mmol/L	1.205	1.07	1.48	1.212 ± 0.091	1.15	1.05	1.44	1.168 ± 0.095
Hct	%	54	5.3	63	51.652 ± 10.95	48	32	59	48.538 ± 5.907
SO2	%	97	37	99	90.346 ± 15.43	100	92	100	99.538 ± 1.606
THBC	g/L	184	133	214	182.16 ± 17.58	163	109	201	165.077 ± 20.112

**Table 3 animals-13-01162-t003:** Mean, standard deviation (SD), median, and range (Min–Max) of blood gas and lactate data provided in awake (T1) and isoflurane anaesthesia (T2) of Chinese pangolins (n_adult_ = 28).

Item	Unit	Awake				Isoflurane Anaesthesia			
		Median	Min	Max	Mean ± SD	Median	Min	Max	Mean ± SD
Body temperature	°C	32.75	31.9	33.8	32.739 ± 0.479	33.1	32.2	34.7	33.296 ± 0.661
H^+^	10^−7^ mol/L	5.248	3.311	7.079	5.301 ± 0.999	5.070	4.266	10.715	5.604 ± 1.368
pH	—	7.28	7.15	7.48	7.284 ± 0.086	7.295	6.97	7.37	7.262 ± 0.092
PO_2_	mmHg	94	25	165	87.929 ± 34.333	296	64	456	280.357 ± 112.034
PCO_2_	mmHg	70.5	44	95	69.036 ± 12.926	81.5	58	109	81.536 ± 11.403
pHt	—	7.34	7.21	7.53	7.343 ± 0.088	7.35	7.02	7.43	7.314 ± 0.095
PO_2_t	mmHg	72	18	148	68.929 ± 30.262	279.5	50	431	262.143 ± 110.608
PCO_2_t	mmHg	58	36	78	57.286 ± 10.561	69	51	91	69.286 ± 9.63
TCO_2_	mmol/L	35	24	48.8	34.793 ± 6.2	39.2	28.4	48.7	39.218 ± 4.898
Beecf	mmol/L	5.25	−6	21.7	5.625 ± 6.827	10.05	−6.7	20.9	9.671 ± 6.112
BE(B)	mmol/L	3.5	−7.4	15.1	2.757 ± 5.838	6.75	−11	14.9	5.668 ± 5.879
HCO_3_^−^	mmol/L	32.95	22.2	46.5	32.668 ± 6.013	36.8	25.1	46.2	36.721 ± 4.932
HCO_3_^−^(std)	mmol/L	27.5	19.1	36.7	26.718 ± 4.571	30.9	16.4	36.6	29.446 ± 4.630
K^+^	mmol/L	4.9	3.7	6.5	4.979 ± 0.597	3.9	3.1	4.7	3.868 ± 0.322
Na^+^	mmol/L	146	139	155	146.571 ± 3.584	145	139	152	144.429 ± 3.426
iCa2^+^	mmol/L	1.25	1.12	1.54	1.266 ± 0.083	1.22	1.07	1.51	1.232 ± 0.095
Glu	mmol/L	4.9	2.6	7.2	5.082 ± 0.926	5.6	4.2	7.5	5.768 ± 0.996
Lac	mmol/L	7.7	3.2	14.6	8.124 ± 3.739	4.45	1.3	14.9	5.386 ± 3.615
nCa^2+^	mmol/L	1.18	1.1	1.47	1.208 ± 0.094	1.15	0.96	1.45	1.166 ± 0.102
Hct	%	59	49	64	58.200 ± 4.5	53	38	60	50.821 ± 6.622
SO2	%	96.5	31	100	88.179 ± 17.889	100	84	100	98.750 ± 3.555
THBC	g/L	201	167	221	198.808 ± 15.654	180	128	204	172.714 ± 22.594

## Data Availability

The all data presented in the study are available under reasonable request from the corresponding author.

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
