# Peer review of "Comparison of Venous Blood Gas and Biochemical Parameters in Sunda Pangolin (Manis javanica) and Chinese Pangolin (Manis pentadactyla) before and after Isoflurane Anesthesia"

_animals, 2023, doi:10.3390/ani13071162_

Round 1
Reviewer 1 Report
The manuscript presents relevant and new information about both species. The availability of these data and the establishment of reference values can help clinicians who focus on rehabilitating these animals.
Despite the merits of the authors, the manuscript needs a careful review of English, some paragraphs are confusing, and some ideas may be poorly conveyed.
The bibliographic review needs to be improved, there are miscited references and formatting errors that need to be corrected.
Authors should refer if the animals were anesthetized only for this study or if the study was performed during an anesthesia episode for other procedures.
The statistical analysis proposed in the materials and methods is suitable for the study in question, but some of these results are not exposed or discussed.
Please read my comments and suggestions.
Comments
L17 this study aims/pretend to analyze...
L 19 Reference ranges... were
L20 rewrite
L22 Partial pressure of oxygen
L27-28 rewrite.
L29 rewrite No significant differences in mean pH or blood ionized calcium (iCa) were observed during anesthesia.
L30 Is this important? If you are delivering reference ranges for each species why the need for comparison between them?
L23-33 rewrite.
L41 Did not find any sentence in reference 3 corroborating its use here
L44 This sentence is not finished.
L47 I don ́t think that adding bibliographic references here is necessary
L51 Rehabilitation centers is the preferred term.
L53-54 Review writing
L61 This reference can ́t be applied here, I think you meant 29
L65 rewrite
L67 gases
L78 This reference does not support the sentence
L92 Coccygeal vein? Tail ventral vein
L93 flow of oxygen/min
L95 same as before
L99 was
L102 Why did you chose a 10m interval between samples. Is this time lapse representative of an anesthetic period? The animals were only anesthetized for the purpose of making this research?
L103 were
L102 to 106 You can fuse this sentences and avoid repeating the blood collection in each sentence
L151-152 rewrite, confusing statement
L153-154 Do you need to compare the values between the two species? Where’s that comparison in the text?
L155-160 rewrite, confusing statements
L182 Wrong citation
L184-185 Lacking a bibliographic reference.
L186-187 Do you mean by throat the larynx? If so, we should expect low SPO2, PO2 and a high
PCO2. If you did had cases like this, I think it should be mentioned and how it was corrected since the animals weren’t intubated.
L192 How long will this autocompensation mechanism endure. The measurements were made 10m after anesthesia induction. We can ́t assume that it would be able to cope with this high
pCO2 levels for a long time.
L198-199 The reference used in this sentence as reference to another author, please revisit the right author for this citation
L203 H+ concentrations aren ́t available in the tables
L206 redundant
L211 and 213 Wrong citation
L215 enter citation
L217 Huong?
L222 wrong citation
L226 gases
Results
On lines L122 & L123 and L126 & L127 it is described all the statistical analysis done with the data. There ́s no references to this data in the results section.
Table 2 Please review your data for iCa2+ in T2
Author Response
Reviewer 1
Despite the merits of the authors, the manuscript needs a careful review of English, some paragraphs are confusing, and some ideas may be poorly conveyed.
>>>>>>Response: Thank you very much for the constructive feedback. We revised and polished the article, and removed some unnecessary repetition, hope it will suit for article publication.
The bibliographic review needs to be improved, there are miscited references and formatting errors that need to be corrected.
>>>Response: Thanks to your suggestions, we have corrected the miscited references and formatting errors.
Authors should refer if the animals were anesthetized only for this study or if the study was performed during an anesthesia episode for other procedures.
>>>Response: The anesthesia of the pangolin was not solely for this study. Rather, before releasing pangolins into the wild, we need to anesthetize pangolins to facilitate a more comprehensive physical examination, such as B-ultrasonography and X-ray inspection. Hematological testing is only one of the items on the physical examination.
The statistical analysis proposed in the materials and methods is suitable for the study in question, but some of these results are not exposed or discussed.
>>>Response: We add the experimental results and describe them in detail. Please see lines: 145-157.
Detailed comments
Line 17: this study aims/pretend to analyze...
>>>Response: Revised.
Line 19: Reference ranges... were
>>>Response: Revised.
Line 20: rewrite
>>>Response: We have rewritten this sentence as your suggestion. Change to “The results obtained showed that the blood gas index trends of the two pangolin species before and after isoflurane anesthesia were the same”. Please see lines: 28-30.
Line 22: Partial pressure of oxygen
>>>Response: Replaced.
Line 27-28: rewrite.
>>>Response: We have rewritten this sentence as your suggestion. Please see lines: 33-35.
Line 29: rewrite No significant differences in mean pH or blood ionized calcium (iCa) were observed during anesthesia.
>>>Response: Replaced.
Line 30: Is this important? If you are delivering reference ranges for each species why the need for comparison between them?
>>>Response: Important. This is one of the only reports describing the blood gas values of two species of pangolins distributed in Asia under isoflurane anesthesia. Although no studies have reported base-line blood gas references between two or more pangolin populations, this study provides a reference for future comparisons of differences in blood gas values between pangolin species.
Line 23-33: rewrite.
>>>Response: We have rewritten this sentence as your suggestion. Please see lines: 30-36.
Line 41: Did not find any sentence in reference 3 corroborating its use here.
>>>Response: Thanks to your suggestions, we have corrected the miscited references and formatting errors. Please see lines: 42.
Line 44: This sentence is not finished.
>>>Response: We have deleted “of their”, and changed it to “a low reproductive rate”.
Line 47: I don ́t think that adding bibliographic references here is necessary.
>>>Response: We believe that the addition of literature here is to show the reader which countries or regions have carried out pangolin rescue work, and that this is supported by the literature. On the one hand, the reader can see that these countries are currently carrying out the pangolin rescue work mentioned in the literature, and on the other hand, the common difficulties faced by these countries or regions in their pangolin conservation efforts——the lack of understanding of pangolin physiological characteristics and health issues.
Line 51: Rehabilitation centers is the preferred term.
>>>Response: Replaced.
Line 53-54: Review writing.
>>>Response: Revised.
Line 61: This reference can ́t be applied here, I think you meant 29.
>>>Response: Thanks to your suggestions, we have corrected the miscited references and formatting errors.
Line 65: rewrite.
>>>Response: Revised.
Line 67: gases.
>>>Response: Replaced.
Line 78: This reference does not support the sentence.
>>>Response: These textual descriptions are exactly the criteria used in the literature as an assessment of the physical health of rescue pangolins. Please see “Animals” in the “Materials and Methods” section of the references for details.
Figure 1. The section of the literature describing the health of the animal (font marked on yellow background).
Line 92: Coccygeal vein? Tail ventral vein.
>>>Response: Changed to “the tail ventral vein”.
Line 93: flow of oxygen/min.
>>>Response: Revised.
Line 95: same as before.
>>>Response: Revised.
Line 99: was.
>>>Response: Replaced.
Line 102: Why did you chose a 10m interval between samples. Is this time lapse representative of an anesthetic period? The animals were only anesthetized for the purpose of making this research?
>>>Response: The anesthesia of the pangolin was not solely for this study. Rather, before releasing pangolins into the wild, we need to anesthetize pangolins to facilitate a more comprehensive physical examination, such as B-ultrasonography and X-ray inspection. Hematological testing is only one of the items on the physical examination. We chose a 10 min time interval into collecting blood samples, referring to the study of R yan. Anesthesia for 10 min resulted in anesthesia stabilization for all pangolins undergoing anesthesia, as confirmed both in the literature and in the present study.
Line 103: were.
>>>Response: Replaced.
Line 102 to 106: You can fuse this sentences and avoid repeating the blood collection in each sentence.
>>>Response: Revised.
Line 151-152: rewrite, confusing statement.
>>>Response: Changed to “This study provides the first blood gas reference ranges for wild Sunda pangolins (M. javanica) and Chinese pangolins (M. pentadactyla)”.
Line 153-154: Do you need to compare the values between the two species? Where’s that comparison in the text?
>>>Response: We have added this section to the results. Please see lines: 145-157. We believe that comparing the differences in blood gas baseline values between two or more pangolin populations may provide a reference for future comparisons of differences in blood gas values between pangolin species.
Line 155-160: rewrite, confusing statements.
>>>Response: Revised. Please see lines: 163-168.
Line 182: Wrong citation.
>>>Response: Revised.
Line 184-185: Lacking a bibliographic reference.
>>>Response: Added.
Line 186-187: Do you mean by throat the larynx? If so, we should expect low SPO2, PO2 and a high PCO2. If you did had cases like this, I think it should be mentioned and how it was corrected since the animals weren’t intubated.
>>>Response: It’s throat the larynx. After we induce anesthesia, we will pull out all the tongues of the pangolin as much as possible to avoid this situation.
Line 192: How long will this autocompensation mechanism endure. The measurements were made 10m after anesthesia induction. We can ́t assume that it would be able to cope with this high pCO2 levels for a long time.
>>>Response: As you said we did not perform tracheal intubation on the pangolin. For this compensation mechanism, we are not sure how long the pangolin will last at this time, so we try to minimize the duration of continuous anesthesia for the pangolin. But we will overcome the problem of tracheal intubation in pangolins and reduce the potential risk of current anesthesia in the future.
Line 198-199: The reference used in this sentence as reference to another author, please revisit the right author for this citation.
>>>Response: We have corrected the cited references. Please see lines: 210.
Line 203: H+ concentrations aren ́t available in the tables.
>>>Response: added.
Line 206: redundant.
>>>Response: Deleted.
Line 211 and 213: Wrong citation.
>>>Response: Revised.
Line 215: enter citation.
>>>Response: Added.
Line 217: Huong?
>>>Response: Revised.
Line 222: wrong citation.
>>>Response: Revised.
Line 226: gases.
>>>Response: Replaced.
Results
On lines L122 & L123 and L126 & L127 it is described all the statistical analysis done with the data. There ́s no references to this data in the results section.
>>>Response: We have added the results of the statistical analysis to the “Results”. In the revised manuscript, we tried our best to present more details in the results. Please see lines: 145-157.
Table 2 Please review your data for iCa2+ in T2.
>>>Response: Revised.

Reviewer 2 Report
The work provides a new useful knowledge base to be used as a helpful diagnostic and prognostic tool to assess and monitoring pangolin health status also considering the conservation concern of these species.
Methods included samples collection are adequate, results interesting, innovative and well discussed. However, significant results obtained for each analysed species need to be better highlighted both in the results section and abstract.
Moreover, supplementary material provided are not cited in the manuscript, please added.
English needs correcting/improving. I suggest avoiding the use of personal language and replaced with the use of impersonal phrase, passive verb or change of subject. Some examples:
L 83: Change “We measured the total…assess age” as “Total body length and tail length of each pangolin were measured to accurately assess age”.
Other minor comments:
L17: This study aimed to …
L18-20: Additional…were established.
L20-25: Results obtained showed……, mean blood sodium, glucose and lactate concentration, oxygen pressure….pangolins
L20-25: Is not clear what these results indicate, please rewritten
L25-30: Are these results valid for both species analysed? Please specify and highlight it also in the results section
L32: …for these pangolins species.
L44: remove “of their”
L129-130: Data also previously provided in materials and methos section, please remove
L216-218: In the same species of pangolins? Or other? Please specify
Author Response
Reviewer 2
Methods included samples collection are adequate, results interesting, innovative and well discussed. However, significant results obtained for each analysed species need to be better highlighted both in the results section and abstract.
Moreover, supplementary material provided are not cited in the manuscript, please added.
>>>>>>Response: We thank the reviewer for the insightful criticism. We have added references to supplementary material in the article.
English needs correcting/improving. I suggest avoiding the use of personal language and replaced with the use of impersonal phrase, passive verb or change of subject. Some examples:
L 83: Change “We measured the total…assess age” as “Total body length and tail length of each pangolin were measured to accurately assess age”.
>>>>>>Response: Thank you very much for the constructive feedback. We revised and polished the article, hope it will suit for article publication. We have replaced the “We measured the total…assess age” as “Total body length and tail length of each pangolin were measured to accurately assess age”. Please see lines: 85-86.
Detailed comments
Lines 17: This study aimed to …
>>>Response: Changed.
Lines 18-20: Additional…were established.
>>>Response: Changed.
Lines 20-25: Results obtained showed……, mean blood sodium, glucose and lactate concentration, oxygen pressure….pangolins
>>>Response: Changed.
Lines 20-25: Is not clear what these results indicate, please rewritten.
>>>Response: We thank the reviewer for the insightful criticism. We have rewritten the sentence. Please see lines: 28-33.
Lines 25-30: Are these results valid for both species analysed? Please specify and highlight it also in the results section.
>>>Response: These results, which are valid for both species analyzed and are described in our results section.
Lines 32: …for these pangolins species.
>>>Response: Changed.
Lines 44: remove “of their”.
>>>Response: Deleted.
Lines 129-130: Data also previously provided in materials and methos section, please remove.
>>>Response: Deleted.
Lines 216-218: In the same species of pangolins? Or other? Please specify.
>>>Response: This refers to the Sudan pangolins and the Chinese pangolins. Please see lines: 227-229.

Round 2
Reviewer 1 Report
The authors made a good revision and the manuscript needs only minor modifications before being eligible for publication. Please see my comments on the revised version.
Comments
The authors made a good revision of the manuscript, I leave here some more comments.
Line 74 “ from different counties in Chinese provinces that were transported to the Guangdong Provincial Wildlife…”
Line 78 Thank you for replying, it was my mistake.
Line 145 Statistical correlations
Line 147 remove partial pressures
Line 154 enter the p value for the parameters that were higher between the two species
Line 195 remove throat and substitute throat for larynx in the next lines
Lines 196 to 199 I think that you can rewrite this sentence and end it suggesting pulling out the tongue of anesthetized pangolins as a preventive measure (an endoscope may not be available in every anesthetic episodes)
Line 228 Jennifer et al (date), ahmad et al (date) Khatri-Chhetri et al (date)
Conclusion
Line 257 to 261 Your work will be the first reference to the normal values for blood gases in this two species of Pangolins. I do think this should be stated in the conclusion as you had it before. I would add a reference to the fact that these values are from rehabilitated healthy pangolins and although probably won’t reflect the values of dehydrated and starving pangolins it will help clinicians treating animals as they now what values to expect in healthy individuals.

Author Response
Reviewer Comments:
Reviewer 1
Detailed comments
Line 74: “from different counties in Chinese provinces that were transported to the Guangdong Provincial Wildlife…”
>>>Response: Changed.
Line 78: Thank you for replying, it was my mistake.
>>>Response: We also appreciate your comments and suggestions on the manuscript.
Line 145: Statistical correlations.
>>>Response: Changed.
Line 147: remove partial pressures.
>>>Response: Deleted.
Line 154: enter the p value for the parameters that were higher between the two species.
>>>Response: Thank you for your careful reading of our manuscript and give us a
significant reminding. We ignore one important issue: the two pangolin themselves species may have differences in blood gas values. Meanwhile, there may be factors such as different food habits, living habits, and places of origin between the two animals that can affect the difference in blood gas values between species. Although comparing blood gas values between two pangolin species is significantly different, it is not statistically significant, so we decided to delete this sentence.
Line 195: remove throat and substitute throat for larynx in the next lines.
>>>Response: Thanks for your valuable counsel. We have changed this sentence by
following your suggestion. Please see lines: 195.
Line 196 to 199: I think that you can rewrite this sentence and end it suggesting pulling out the tongue of anesthetized pangolins as a preventive measure (an endoscope may not be available in every anesthetic episodes).
>>>Response: Thanks for your valuable counsel. We have changed this sentence by
following your suggestion. Please see lines: 196-200.
Line 228: Jennifer et al (date), ahmad et al (date) Khatri-Chhetri et al (date).
>>>Response: Added.
Conclusion
Line 257 to 261 Your work will be the first reference to the normal values for blood gases in this two species of Pangolins. I do think this should be stated in the conclusion as you had it before. I would add a reference to the fact that these values are from rehabilitated healthy pangolins and although probably won’t reflect the values of dehydrated and starving pangolins it will help clinicians treating animals as they now what values to expect in healthy individuals..
>>>Response: Thank you for your suggestion, and we have added that the value of this study is to provide clinicians with a reference range of blood gases for the treatment of pangolins. Please see lines: 264-267.
